# A cross–sectional study on the prevalence and associated risk factors for workplace violence against Chinese nurses

Lei Shi,[1] Danyang Zhang,[2] Chenyu Zhou,[1] Libin Yang,[3] Tao Sun,[1] Tianjun Hao,[4] Xiangwen Peng,[2] Lei Gao,[1] Wenhui Liu,[3] Yi Mu,[5] Yuzhen Han,[6] Lihua Fan[1]

► Prepublication history and additional material are available. To view these files please visit the journal online (http://dx.doi.org/10.1136/bmjopen-2016-013105).

[1]Department of Health Management, School of Public Health, Harbin Medical University, Harbin, China
[2]Operating Section, The First Affiliated Hospital of Harbin Medical University, Harbin, China
[3]Department of Medical Education, School of Public Health, Harbin Medical University, Harbin, China
[4]Department of Scientific Research, The Second Affiliated Hospital of Harbin Medical University, Harbin, China
[5]Department of Customer Service, Beijing Children's Hospital, Capital Medical University, Beijing, China
[6]Department of Medical Disputes, The Fourth Affiliated Hospital of Harbin Medical University, Harbin, China

**Correspondence to**
Professor Lihua Fan;
lihuafan@126.com

## ABSTRACT

**Objectives** The purpose of the present study was to explore the characteristics of workplace violence that Chinese nurses at tertiary and county–level hospitals encountered in the 12 months from December 2014 to January 2016, to identify and analyse risk factors for workplace violence, and to establish the basis for future preventive strategies.

**Design** A cross–sectional study.

**Setting** A total of 44 tertiary hospitals and 90 county–level hospitals in 16 provinces (municipalities or autonomous regions) in China.

**Methods** We used stratified random sampling to collect data from December 2014 to January 2016. We distributed 21 360 questionnaires, and 15 970 participants provided valid data (effective response rate=74.77%). We conducted binary logistic regression analyses on the risk factors for workplace violence among the nurses in our sample and analysed the reasons for aggression.

**Results** The prevalence of workplace violence was 65.8%; of this, 64.9% was verbal violence, and physical violence and sexual harassment accounted for 11.8% and 3.9%, respectively. Frequent workplace violence occurred primarily in emergency and paediatric departments. Respondents reported that patients' relatives were the main perpetrators in tertiary and county–level hospitals. Logistic regression analysis showed that respondents' age, department, years of experience and direct contact with patients were common risk factors at different levels of hospitals.

**Conclusions** Workplace violence is frequent in China's tertiary and county–level hospitals; its occurrence is especially frequent in the emergency and paediatric departments. It is necessary to cope with workplace violence by developing effective control strategies at individual, hospital and national levels.

## BACKGROUND

Workplace violence (WPV) towards health service professionals is recognised as a global public health issue, and it has attracted worldwide attention.[1–4] Previous studies have suggested that health professionals have a higher risk of experiencing WPV than any other professionals.[2 3] Further, the incidence

### Strengths and limitations of this study

► We used a large, nurse-based sample in our study.
► Sample selection was reasonable and representative, and was distributed in the eastern, middle and western regions of China.
► Our study compared differences between nurses who had experienced workplace violence in comprehensive public tertiary and county–level hospitals.
► The study provides the basis for establishing operational guidelines for preventing workplace violence in China.
► The retrospective approach to self-reported workplace violence used by respondents may cause recall bias.

rate of WPV differs among nurses in various countries; for instance, the incidence was 76.0% in Greece, 82% in Pakistan and 67% in Italy.[5 6]

Several substantial studies have suggested that nurses have a high risk of experiencing WPV.[6–10] During the past 12 months, the incidence rate of physical violence for nurses in Ethiopia,[7] South Korea,[8] Jordan,[9] Germany[10] and Iran[3] ranged from 18.22% to 56.0%, the verbal abuse rate being from 63.8% to 89.58% and the sexual harassment rate from 4.7% to 19.7%. WPV occurs primarily in the emergency wards and psychiatric departments of hospitals.[11 12] Research into the experience of nurses in these departments in the USA,[13] Switzerland[14] and Jordan[15] has demonstrated that they experience a higher incidence of WPV than do nurses in other departments.

In China, WPV in hospitals has increased gradually over the past few decades. According to the report from the Chinese Hospital Association, the proportion of hospitals experiencing WPV increased from 90% in 2008 to

96% in 2012, and the prevalence of sexual harassment has increased year by year.[16]

Most previous studies of WPV against nurses in China have been conducted in the provinces, and the samples are not sufficiently representative: they do not present an accurate picture of the incidence of WPV against nurses in Chinese general hospitals.[17–19] In this study, WPV was divided into physical violence and psychological violence, which includes verbal violence and sexual harassment in accordance with the definition of WPV in hospitals used by the WHO, the International Labour Office (ILO) and the specific situation in China.

Although violent incidents in hospitals directly harm healthcare workers and hospitals, the ultimate victims are the patients.[20] WPV affects the normal functioning and reputation of hospitals and threatens the personal safety of healthcare workers and patients.[21] Moreover, violent incidents have a negative impact on the psychological welfare of healthcare workers[22] who do not want their children to be engaged in healthcare.[23]

The purpose of the present study was to explore the distribution and characteristics of WPV experienced by Chinese nurses at tertiary and county–level hospitals in the 12 months from December 2014 to January 2016, to identify and analyse the risk factors for WPV, thus providing a basis for future preventive strategies.

## METHODS
### Sample and procedure
A cross–sectional survey was designed based on the geographical location and level of economic development in the eastern (Beijing, Tianjin, Hebei, Shandong, Guangdong, Liaoning), middle (Shanxi, Henan, Anhui, Hunan, Heilongjiang) and western (Ningxia, Shannxi, Gansu, Sichuan, Chongqing) regions of China. In 2015, there were 3069 general and public hospitals in China and approximately 920 700 registered nurses in public tertiary and county–level hospitals. We selected a sample of 21 360, approximately 2.30% of all nurses. In order to select the same proportion of the workforce from each department, we sought to recruit 120 nurses from the departments of internal medicine and surgery, 80 nurses from the departments of emergency, neurology, obstetrics and gynaecology, and paediatrics, and 40 nurses from the departments of stomatology, ophthalmology, and ear, nose and throat in each tertiary hospital. We distributed 10 560 questionnaires to 44 tertiary hospitals. We sought to recruit nurses at county–level hospitals in the same proportions: 60 from the departments of internal medicine and surgery, 40 from the departments of emergency, neurology, obstetrics and gynaecology, and paediatrics, and 20 from the departments of stomatology, ophthalmology, and ear, nose and throat at each county–level hospital. We distributed 10 800 questionnaires to the 90 county–level hospitals.

All 134 hospitals agreed to participate in the study. The inclusion criteria for participants were voluntary participation by nurses engaged in clinical work who had at least 1 year of professional experience in hospitals.

The survey was conducted from December 2014 to January 2016. We obtained permission from the managers and human resources departments of the hospitals. The researchers used a stratified random sampling method to collect the data. Each participant had 2 days to complete the self-administered questionnaire. Collected data were kept confidential and used only for academic research. We distributed 21 360 questionnaires and received 15 970 valid questionnaires (effective response rate=74.77%).

### Questionnaire
We developed the questionnaire based on three documents. First, we used the revised Survey of Violence Experienced by Staff (SOVES-G-R) developed by Hahn *et al* to formulate the questionnaire.[24] The revised SOVES-G-R was used to measure perpetrators' characteristics and the way to deal with violence for nurses. Next, we used items from the Chinese version of the Workplace Violence Scale in the literature (the frequency of violence, different levels of the severity of physical violence, and hospital attitudes to WPV and intervention strategies) to develop the questionnaire.[25] Chen and Wang's study incorporated the levels of the severity of physical violence, and hospital attitudes to WPV and intervention strategies. In addition, we used the questionnaires from the ILO, ICN (International Council of Nurses), WHO and PSI (Public Services International) joint programme for measuring WPV (eg, types of WPV, time and place the violence occurred, and perceptions and attitudes of nurses towards WPV).[26]

We pre-tested the questionnaire with 367 nurses from three public tertiary and two public county–level hospitals in Heilongjiang Province and revised the questionnaire after the test. Further, we invited 18 healthcare related experts to assess the accuracy, comprehensiveness and sensitivity of the items in the questionnaire. In order to ensure the reliability of the revised questionnaire according to expert opinion, we selected 835 nurses from five public tertiary and four public county–level hospitals in Heilongjiang Province to measure the reliability of the questionnaire again. We measured the questionnaire's reliability twice using SPSS19.0; Cronbach's alpha coefficient was 0.84 and 0.86, respectively.

### Data analysis
EpiData V.3.1 was used to establish the study's database. To ensure accuracy, the data were checked by trained personnel after all surveys were completed and entered. We used IBM SPSS19.0 and Excel for statistical analysis of the relevant quantitative data. Univariate analysis and $\chi^2$ tests ($\chi^2$ tests were used to analyse the relationship between the demographic characteristics of respondents and the incidence rate of WPV). We employed logistic regression analysis to test the relationship between nurses experiencing hospital WPV as the dependent variable, and the variables with statistical and clinical practice significance

**Table 1** Participants' demographic characteristics (n=15 970)

| Characteristic | Tertiary hospitals (n=9142) | | County–level hospitals (n=6828) | | Total (n=15 970) | |
|---|---|---|---|---|---|---|
| | n | % | n | % | n | % |
| **Gender** | | | | | | |
| Male | 234 | 2.6 | 143 | 2.1 | 377 | 2.4 |
| Female | 8908 | 97.4 | 6685 | 97.9 | 15 593 | 97.6 |
| **Age (years)** | | | | | | |
| ≤30 | 5131 | 56.1 | 4033 | 59.1 | 9164 | 57.4 |
| 31–50 | 3785 | 41.4 | 2611 | 38.2 | 6396 | 40.0 |
| ≥51 | 226 | 2.5 | 184 | 2.7 | 410 | 2.6 |
| **Educational level** | | | | | | |
| Below undergraduate | 3129 | 34.2 | 4161 | 60.9 | 7290 | 45.6 |
| Undergraduate | 5910 | 64.6 | 2663 | 39.0 | 8573 | 53.7 |
| Master's or above | 103 | 1.1 | 4 | 0.1 | 107 | 0.7 |
| **Marital status** | | | | | | |
| Married | 5678 | 62.1 | 4293 | 62.9 | 9971 | 62.4 |
| Unmarried | 3342 | 36.6 | 2415 | 35.4 | 5757 | 36.1 |
| Divorced or widowed | 122 | 1.3 | 120 | 1.7 | 242 | 1.5 |
| **Professional title** | | | | | | |
| Junior | 6249 | 68.4 | 4845 | 71.0 | 11 094 | 69.5 |
| Intermediate | 2242 | 24.5 | 1587 | 23.2 | 3829 | 24.0 |
| Senior | 651 | 7.1 | 396 | 5.8 | 1047 | 6.5 |
| **Employment form** | | | | | | |
| Regular staff | 5077 | 55.5 | 3339 | 48.9 | 8416 | 52.7 |
| Temporary employee | 4065 | 44.5 | 3489 | 51.1 | 7554 | 47.3 |
| **Average monthly income (RMB)** | | | | | | |
| ≤3000 | 3747 | 41.0 | 4916 | 72.0 | 8663 | 54.3 |
| 3000–5000 | 4495 | 49.2 | 1867 | 27.3 | 6362 | 39.8 |
| 5000–10 000 | 872 | 9.5 | 42 | 0.6 | 914 | 5.7 |
| >10 000 | 28 | 0.3 | 3 | 0.1 | 31 | 0.2 |
| **Department** | | | | | | |
| Emergency | 514 | 5.6 | 522 | 7.6 | 1036 | 6.5 |
| Internal medicine | 2850 | 31.2 | 1962 | 28.7 | 4812 | 30.1 |
| Surgery | 1903 | 20.8 | 1386 | 20.3 | 3289 | 20.6 |
| Gynaecology and obstetrics | 426 | 4.7 | 610 | 8.9 | 1036 | 6.5 |
| Paediatrics | 516 | 5.6 | 627 | 9.2 | 1143 | 7.2 |
| Other | 2933 | 32.1 | 1721 | 25.3 | 4654 | 29.1 |
| **Years of experience** | | | | | | |
| 1–4 | 3511 | 38.4 | 3124 | 45.7 | 6635 | 41.5 |
| 5–10 | 2803 | 30.7 | 1597 | 23.4 | 4400 | 27.6 |
| 11–20 | 1534 | 16.8 | 1099 | 16.1 | 2633 | 16.5 |
| ≥21 | 1294 | 14.1 | 1008 | 14.8 | 2302 | 14.4 |
| **Working time** | | | | | | |
| 0–2 hours | 164 | 1.8 | 52 | 0.8 | 216 | 1.3 |
| 2–4 hours | 252 | 2.8 | 120 | 1.8 | 372 | 2.3 |
| 4–6 hours | 264 | 2.9 | 113 | 1.6 | 377 | 2.4 |
| 6–8 hours | 3713 | 40.6 | 3087 | 45.2 | 6800 | 42.6 |
| >8 hours | 4749 | 51.9 | 3456 | 50.6 | 8205 | 51.4 |
| **Direct contact with patients** | | | | | | |
| 0–2 hours | 180 | 2.0 | 106 | 1.6 | 286 | 1.8 |
| 2–4 hours | 237 | 2.6 | 251 | 3.7 | 488 | 3.0 |
| 4–6 hours | 747 | 8.2 | 575 | 8.4 | 1322 | 8.3 |
| 6–8 hours | 7978 | 87.2 | 5896 | 86.3 | 13 874 | 86.9 |

**Table 2** Incidence (%) of exposure to workplace violence

| | Tertiary hospitals | | | | | | County–level hospitals | | | | | |
| | Physical violence | | Verbal violence | | Sexual harassment | | Physical violence | | Verbal violence | | Sexual harassment | |
| Type | n | % | n | % | n | % | n | % | n | % | n | % |
| | 1047 | 11.5 | 5875 | 64.3 | 402 | 4.4 | 845 | 12.4 | 4494 | 65.8 | 228 | 3.3 |

as the covariates. We conducted binary logistic regression analyses on the risk factors for WPV among the nurses in our sample. OR and 95% CI were calculated; α=0.05 was the test and p<0.05 was considered statistically significant.

### Ethics approval
Ethics approval to undertake this study was granted by the research ethics committee of Harbin Medical University in March 2014. We obtained consent from each hospital involved in the research processes. All participants gave informed consent to the researchers or to their head nurses before the survey, and participants' personal information was kept confidential.

## RESULTS
### Participants' demographic characteristics
We received 17 865 responses, and 15 970 respondents met our inclusion criteria. Most respondents had completed undergraduate studies and were <30 years old; however, the number of female respondents was significantly higher compared with male respondents (table 1).

### Incidence of exposure to WPV during the past 12 months
In the 12–month survey period, 10 502 nurses experienced WPV. Total prevalence was 65.8%, of which verbal violence accounted for 64.9% (10369/15970), physical violence for 11.8% (1892/15970) and sexual harassment for 3.9% (630/15970). The incidence rate of WPV in the different types of hospitals is shown in table 2.

### Prevalence of WPV for different demographic variables
Nurses with frequent contact with patients were more likely to experience WPV in both tertiary and county–level hospitals (table 3). There was a significant difference in the incidence rate of WPV among different professional titles for nurses in the tertiary hospitals ($\chi^2$ = 6.6, p<0.05). In contrast, there was no significant difference in county–level hospitals ($\chi^2$ = 1.5, p>0.05).

### Characteristics of perpetrators and victims' responses
Patients' relatives were the main perpetrators of WPV in both tertiary (83.1%) and county–level hospitals (85.2%). Most attacks on nurses occurred in the wards and during the day shift. More than 60% of the victims responded with tolerance, patience and understanding (table 4).

### Risk factors associated with WPV against nurses
Participants' age, department, years of experience and length of time in direct contact with patients were risk factors for WPV against nurses. Binary logistic regression analysis results demonstrated clearly that <8 hours'

working time is a protective factor against WPV for nurses in county–level hospitals. Nurses in emergency departments were almost three times (OR=2.993, 95% CI 2.364 to 3.789) more likely to experience WPV in tertiary hospitals, and 3.387 times (95% CI 2.648 to 4.332) more likely in county–level hospitals than nurses in other departments. Surprisingly, nurses with 5–10 years' of experience were at the highest risk of WPV in county– level hospitals, while those with 11–20 years' experience were at the highest risk in tertiary hospitals. Further, our results showed that the likelihood of WPV rises with the length of treatment time in county–level hospitals (table 5).

## DISCUSSION
Previous studies have found that WPV exists in all hospitals but differs in some aspects.[27–31] The total incidence rate of WPV was 65.8% over the 12–month period. This is slightly higher than the figure of 64.48% reported for Guangzhou.[32] Compared with previous Chinese studies,[17–19] our study provides a comprehensive depiction of the incidence of WPV in Chinese comprehensive public hospitals. We have also documented the characteristics of the perpetrators and the coping style of nurses who experience WPV. This study demonstrated that WPV in China is higher than in other countries.[33–36] Moreover, we found that WPV towards nurses was frequent, including verbal violence (64.9%), physical violence (11.8%) and sexual harassment (3.9%). This may be related to the fact that nurses are frequently in direct contact with patients and their relatives in their daily work. The findings also showed that nurses in county–level hospitals are more likely to experience WPV than nurses in tertiary hospitals. This may be related to the level of education of patients and their relatives in county–level hospitals. In addition, the significant agricultural population in China is more likely to receive medical treatment at county-level hospitals, increasing the likelihood of WPV. In short, nurses have a high risk of experiencing WPV in the healthcare sector.[37–39]

Our survey also revealed that respondents aged 30 years or younger have a higher risk of WPV than their older colleagues. This phenomenon may be attributable to the fact that they have not been nurses for very long, and they lack work experience and communication skills. If that is the case, they need to practice their skills to decrease the odds of making mistakes. Alternatively, because most of the nurses in this age group were single children, they are quick to anger when criticised by patients and their relatives, increasing the possibility of WPV. In contrast, nurses

**Table 3** Characteristics and frequency distributions for workplace violence

| Characteristic | Tertiary hospitals (n=9142) | | | | County–level hospitals (n=6828) | | | |
|---|---|---|---|---|---|---|---|---|
| | n | % | $\chi^2$ | p | n | % | $\chi^2$ | p |
| **Gender** | | | | | | | | |
| Male | 143 | 61.1 | 1.8 | 0.181 | 85 | 59.4 | 3.2 | 0.072 |
| Female | 5820 | 65.3 | | | 4454 | 66.6 | | |
| **Age (years)** | | | | | | | | |
| ≤30 | 3283 | 64.0 | 15.4 | <0.001 | 2679 | 66.4 | 15.6 | <0.001 |
| 31–50 | 2548 | 67.3 | | | 1762 | 67.5 | | |
| ≥51 | 132 | 58.4 | | | 98 | 53.3 | | |
| **Educational level** | | | | | | | | |
| Below undergraduate | 1958 | 62.6 | 15.7 | <0.001 | 2707 | 65.1 | 9.7 | 0.008 |
| Undergraduate | 3941 | 66.7 | | | 1829 | 68.7 | | |
| Master's or above | 64 | 62.1 | | | 3 | 75.0 | | |
| **Marital status** | | | | | | | | |
| Married | 3813 | 67.2 | 27.8 | <0.001 | 2884 | 67.2 | 3.3 | 0.192 |
| Unmarried | 2065 | 61.8 | | | 1581 | 65.5 | | |
| Divorced or widowed | 85 | 69.7 | | | 74 | 61.7 | | |
| **Professional title** | | | | | | | | |
| Junior | 4022 | 64.4 | 6.6 | 0.037 | 3217 | 66.4 | 1.5 | 0.482 |
| Intermediate | 1607 | 71.7 | | | 1068 | 67.3 | | |
| Senior | 434 | 66.7 | | | 254 | 64.1 | | |
| **Employment form** | | | | | | | | |
| Regular staff | 3290 | 64.8 | 0.9 | 0.341 | 2213 | 66.3 | 0.1 | 0.733 |
| Temporary employee | 2673 | 65.8 | | | 2326 | 66.7 | | |
| **Average monthly income (RMB)** | | | | | | | | |
| ≤3000 | 2418 | 64.5 | 1.4 | 0.710 | 3275 | 66.6 | 0.3 | 0.955 |
| 3000–5000 | 2953 | 65.7 | | | 1233 | 66.0 | | |
| 5000–10000 | 574 | 65.8 | | | 29 | 69.0 | | |
| >10000 | 18 | 64.3 | | | 2 | 66.7 | | |
| **Department** | | | | | | | | |
| Emergency | 418 | 81.3 | 101.2 | <0.001 | 429 | 82.2 | 158.9 | <0.001 |
| Internal medicine | 1889 | 66.3 | | | 1294 | 66.0 | | |
| Surgery | 1266 | 66.5 | | | 955 | 68.9 | | |
| Gynaecology and obstetrics | 284 | 66.7 | | | 391 | 64.1 | | |
| Paediatrics | 351 | 68.0 | | | 483 | 77.0 | | |
| Other | 1755 | 59.8 | | | 987 | 57.4 | | |
| **Years of experience** | | | | | | | | |
| 1–4 | 2141 | 61.0 | 55.0 | <0.001 | 2006 | 64.2 | 35.2 | <0.001 |
| 5–10 | 1920 | 68.5 | | | 1148 | 71.9 | | |
| 11–20 | 1069 | 69.7 | | | 750 | 68.2 | | |
| ≥21 | 833 | 64.4 | | | 635 | 63.0 | | |
| **Working time** | | | | | | | | |
| 0–2 hours | 87 | 53.0 | 26.1 | <0.001 | 32 | 61.5 | 67.9 | <0.001 |
| 2–4 hours | 152 | 60.3 | | | 71 | 59.2 | | |
| 4–6 hours | 165 | 62.5 | | | 53 | 46.9 | | |
| 6–8 hours | 2367 | 63.7 | | | 1941 | 62.9 | | |
| >8 hours | 3192 | 67.2 | | | 2442 | 70.7 | | |
| **Direct contact with patients** | | | | | | | | |
| 0–2 hours | 72 | 40.0 | 69.8 | <0.001 | 52 | 49.1 | 51.7 | <0.001 |
| 2–4 hours | 127 | 53.6 | | | 144 | 57.4 | | |
| 4–6 hours | 474 | 63.5 | | | 330 | 57.4 | | |
| 6–8 hours | 5290 | 66.3 | | | 4013 | 68.1 | | |

**Table 4** Characteristics of perpetrators and victims' responses

| | Tertiary hospitals (n=5963) | | County–level hospitals (n=4539) | |
|---|---|---|---|---|
| | n | % | n | % |
| **Attack time** | | | | |
| Day shift | 3558 | 59.7 | 2920 | 64.3 |
| Night shift | 1402 | 23.5 | 1195 | 26.4 |
| After work | 1003 | 16.8 | 424 | 9.3 |
| **Attack site** | | | | |
| Outpatient clinic | 388 | 6.5 | 351 | 7.7 |
| Ward | 2754 | 46.2 | 1933 | 42.6 |
| Doctor's office | 198 | 3.3 | 251 | 5.5 |
| Nurse's office or station | 1372 | 23.0 | 1291 | 28.5 |
| Treatment room | 127 | 2.1 | 162 | 3.6 |
| Other | 1124 | 18.9 | 551 | 12.1 |
| **When the violent incident took place** | | | | |
| All alone | 1698 | 28.5 | 1319 | 29.1 |
| Other colleagues on the scene | 4265 | 71.5 | 3220 | 70.9 |
| **Perpetrators*** | | | | |
| Patients | 1784 | 35.9 | 1212 | 26.7 |
| Patients' relatives | 4131 | 83.1 | 3862 | 85.2 |
| Visitors | 748 | 15.0 | 573 | 12.6 |
| Other | 99 | 2.0 | 96 | 2.1 |
| **Gender of the perpetrators*** | | | | |
| Male | 3988 | 81.2 | 3380 | 83.1 |
| Female | 2461 | 50.1 | 1917 | 47.1 |
| **Age group of the perpetrators (years)*** | | | | |
| ≤20 | 258 | 5.1 | 258 | 5.7 |
| 21–30 | 1454 | 28.8 | 1624 | 35.9 |
| 31–40 | 2693 | 53.3 | 2263 | 50.0 |
| 41–50 | 2220 | 44.0 | 1529 | 33.8 |
| 51–60 | 917 | 18.2 | 1011 | 22.3 |
| ≥61 | 445 | 8.8 | 668 | 14.8 |
| **Behavioural response to WPV *** | | | | |
| Tolerance and avoidance | 3167 | 64.1 | 2601 | 63.3 |
| Patience and understanding | 2748 | 55.7 | 2415 | 58.8 |
| Give tit for tat | 80 | 1.6 | 36 | 0.9 |
| Try to explain before resorting to force | 438 | 8.9 | 310 | 7.5 |
| Ask colleagues for help | 940 | 19.0 | 537 | 13.1 |
| Turn to the managers or security staff for help | 1899 | 38.5 | 1334 | 32.5 |
| Ask for help from other patients and relatives | 287 | 5.8 | 165 | 4.0 |
| Call the police | 754 | 15.3 | 573 | 14.0 |
| Other | 166 | 3.4 | 72 | 1.8 |

*Represents multiple choice.

aged 51 years or older had a lower risk of WPV. This may be related to inherent respect for the elderly, a traditional virtue of the Chinese nation for thousands of years.[40]

As shown in our survey, the risk of WPV was not the same in different departments. Substantial previous studies have found that emergency departments are likely to have the highest incidence of WPV, followed by paediatric departments.[24 41] This increased risk may be because emergency departments deal with the most serious patients in complex situations, such as traffic accidents, food poisoning and patients with alcoholism. Further, patients' relatives may be very worried, and if nurses do not share information with patients and if the nurses' communication skills are inadequate, unnecessary conflicts may emerge. Paediatrics is another specialist department with a high risk of WPV. In China, all patients <14 years of age are treated in paediatrics departments. The age range is different in countries around the world. The high risk of WPV in paediatric departments may be because patients are likely to be their parents' only child, and to have been overindulged. This may make nurses' work more difficult and increase the possibility of WPV. Therefore, hospitals and patients should make joint efforts to reduce violent incidents.

**Table 5** Risk factors associated with workplace violence against nurses in hospitals: binary logistic results[*]

| Variable name | | Tertiary hospitals (n=9142) | | | County–level hospitals (n=6828) | | |
|---|---|---|---|---|---|---|---|
| | | Adjusted OR | 95% CI | p Value | Adjusted OR | 95% CI | p Value |
| Age group (years) | ≥51 | 1.0 | Reference | 0.044 | 1.0 | reference | 0.027 |
| | ≤30 | 1.500 | (1.092 to 2.061) | 0.012 | 1.606 | (1.115 to 2.314) | 0.011 |
| | 31–50 | 1.387 | (1.036 to 1.855) | 0.028 | 1.551 | (1.120 to 2.146) | 0.008 |
| Department | Other | 1.0 | Reference | <0.001 | 1.0 | reference | <0.001 |
| | Emergency | 2.993 | (2.364 to 3.789) | <0.001 | 3.387 | (2.648 to 4.332) | <0.001 |
| | Internal medicine | 1.313 | (1.178 to 1.463) | <0.001 | 1.408 | (1.227 to 1.615) | <0.001 |
| | Surgery | 1.341 | (1.187 to 1.514) | <0.001 | 1.644 | (1.414 to 1.912) | <0.001 |
| | Gynaecology and obstetrics | 1.322 | (1.065 to 1.641) | <0.001 | 1.268 | (1.044 to 1.539) | 0.016 |
| | Paediatrics | 1.433 | (1.172 to 1.753) | <0.001 | 2.391 | (1.934 to 2.956) | <0.001 |
| Years of experience | 1–4 | 1.0 | Reference | <0.001 | 1.0 | reference | <0.001 |
| | 5–10 | 1.426 | (1.268 to 1.604) | <0.001 | 1.479 | (1.277 to 1.712) | <0.001 |
| | 11–20 | 1.627 | (1.358 to 1.951) | <0.001 | 1.300 | (1.045 to 1.618) | 0.018 |
| | ≥21 | 1.368 | (1.131 to 1.656) | 0.001 | 1.192 | (0.949 to 1.498) | 0.131 |
| Working time (hours) | >8 hours | | | | 1.0 | reference | <0.001 |
| | 0–2 hours | | | | 0.852 | (0.469 to 1.548) | 0.599 |
| | 2–4 hours | | | | 0.762 | (0.507 to 1.145) | 0.191 |
| | 4–6 hours | | | | 0.492 | (0.329 to 0.735) | 0.001 |
| | 6–8 hours | | | | 0.720 | (0.647 to 0.800) | <0.001 |
| Direct contact with patients (hours) | 0–2 hours | 1.0 | Reference | <0.001 | 1.0 | reference | <0.001 |
| | 2–4 hours | 1.832 | (1.230 to 2.728) | 0.003 | 1.201 | (0.744 to 1.938) | 0.454 |
| | 4–6 hours | 2.722 | (1.939 to 3.819) | <0.001 | 1.220 | (0.785 to 1.895) | 0.377 |
| | 6–8 hours | 3.054 | (2.247 to 4.152) | <0.001 | 1.710 | (1.134 to 2.580) | 0.011 |

*This analysis used data from 44 public tertiary hospitals and 90 public county-level hospitals in China.

On the one hand, hospitals can establish a code green response team, comprising a charge nurse, security personnel and primary nurse to manage any potentially violent situation. Dilman's study demonstrated that 85% of code green calls resulted in successful resolution of the violent incidents.[42] Preventive measures for WPV among nurses include the following: increase awareness of potentially violent patients; wear suitable clothes; maintain proper positioning when communicating with patients; keep a safe distance; maintain the correct posture; and listen actively. A study by Hill et al showed a 65% reduction in staff injuries, from 2.2 per week to 0.77 per week, during the 1 year intervention period.[43] On the other hand, patients and their relatives should show respect to others and understand that nurses' work is complex and professional. With mutual understanding, nurses and patients can help to improve the nurse-patient relationship and achieve harmony. Stievano et al's study has shown that patient care can be affected when nurses are not respected.[44]

This study also demonstrated that years of experience and extended direct contact with patients are risk factors for WPV. Medical treatment processes, waiting time for patients, nurses' attitudes and other aspects of hospital procedure need improvement to enhance patients' satisfaction and reduce conflict. Previous studies have shown that strengthening hospital management can improve patient satisfaction.[45]

Perpetrators' characteristics are equally noteworthy. In the present study, perpetrators were primarily patients' family members, followed by patients. Perpetrators were usually men, aged 31–50 years with an inverted 'U' shape distribution. The ward was the main site of WPV in hospitals, accounting for 46.2% and 42.6% in tertiary and county–level hospitals, respectively. Ward management needs strengthening; for example, surveillance cameras and alarms should be installed in hospital ward corridors, lights should be sufficiently bright in work areas during the night, and so on. Fine safety management can significantly improve the quality of nursing and patients' satisfaction, and reduce conflict.[46]

Although most patients and patients' relatives are well behaved, a few have underhand motives. Such people intentionally disrupt the normal medical order, and even pose a threat to the personal safety of medical staff and other patients, in order to seek monetary compensation. In China, a few years ago, the cost of 'Yi Nao' (where in people employed by those in dispute with the hospital, together with family members of patients, took various measures to put pressure on the hospital, thereby profiteering) was very low. Occasionally, patients and their relatives may use illegal organisations to ask the hospitals for compensation directly, rather than use the normal legal procedures. Hospitals often make concessions by acquiescing to some unreasonable requirements to avoid trouble, preserve their reputation, and ensure normal

medical order. This response has further contributed to the arrogance of those resorting to 'Yi Nao'. Fortunately, 'Yi Nao' was officially criminalised in November 2015. The law now recognises that 'Yi Nao' risks the proper functioning of medical institutions and gives clear legal support to conviction and sentencing. Negative media coverage also had a huge impact on the nurse–patient relationship. Therefore, we suggest that the government should supervise the media to ensure the accuracy and authenticity of media coverage. We further advise hospitals to strengthen the training and management of nurses to reduce the physical and psychological damage to them from WPV. For instance, hospitals could provide violence–related training for nurses and provide post-WPV psychological support or a 'debriefing room,' instruct all staff about the value of nurses and foster nurses' pride in their work and develop an excellent hospital culture. If nurses have the support of managers, they may be more willing to consult psychologists, psychiatrists and mental health professionals to identify and treat any sudden disorder.[47]

We were surprised by victims' coping mechanisms, which were tolerance, avoidance, patience, and under-standing in both tertiary and county–level hospitals. This response may reflect a culture of tolerance toward WPV in China. Most nurses turned to their managers or hospital security personnel for help when they experienced WPV. Thus the organisation's support and care may have significant potential for reducing the harm inflicted on nurses by WPV. Nurses should be familiar with all of the resources in their work and family communities for solving any problems and should keep themselves in good mental health.[47]

The present study has several limitations. First, we collected data about whether nurses had experienced WPV over the previous 12 months, so there may have been recall bias in the results. Second, we studied tertiary and county–level hospitals, but we did not study specialised hospitals. However, this study might be effective in preventing WPV in general and public hospitals.

In general, WPV can be prevented, according to the WHO's report on violence and health.[26] Therefore, to reduce violent incidents, we recommend that tertiary and county–level hospitals develop training against violence that is tailored to their particular situation, as well as the other measures we have suggested above. We also suggest improving psychological resilience for at-risk nurses to mitigate the negative impact of WPV and to prevent chronic diseases and reduce the incidence of mental illness.[48]

## CONCLUSIONS

This study is based on a large sample survey of WPV in tertiary and county–level hospitals in China. The study has demonstrated that there is a high incidence rate of WPV and that occupational safety is insufficient. The frequent occurrence of WPV in emergency and paediatric departments is also remarkable. The incidence of WPV of hospitals is an occupational health hazard and a serious threat to the well being of nurses.

**Acknowledgements** The authors thank the nurses, managers and the Chinese Hospital Association for their assistance and support for this project. The authors also acknowledge Xiaoli Jia and Min Ha for their assistance with the survey.

**Contributors** LS and LF designed the study. DZ, CZ, TH, XP, YM and YH collected the data. LS, LY, TS, LG and WL analysed the data. LS and LF drafted the manuscript. LS, LY and LF revised the manuscript.

**Funding** This study was funded by the National Natural Science Foundation of China, grant No 71473063.

**Competing interests** None declared.

**Ethics approval** The study was approved by the research ethics committee of Harbin Medical University.

**Provenance and peer review** Not commissioned; externally peer reviewed.

**Data sharing statement** No additional data are available.

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
