## [Reviewer comments · BMJ Open]

ARTICLE DETAILS

TITLE (PROVISIONAL)	A cross-sectional study on the prevalence and associated risk factors for workplace violence against Chinese nurses
AUTHORS	Shi, Lei; Zhang, Danyang; Zhou, Chenyu; Yang, Libin; Sun, Tao; Hao, Tianjun; Peng, Xiangwen; Gao, Lei; Liu, Wenhui; Mu, Yi; Han, Yuzhen; Fan, Lihua

VERSION 1 - REVIEW

REVIEWER	Ramacciati Nicola Azienda Ospedaliera di Perugia, Perugia (Italy) Università degli studi di Firenze, Firenze (Italy)
REVIEW RETURNED	22-Jul-2016

GENERAL COMMENTS	Interesting article. Would you like to explain in the main text what's "Yi Nao", please? The study limitations aren't discussed in the paper. Will you add this paragraph, please?
--

REVIEWER	Rula Al-Rimawi Jordan- balqaa applied university
REVIEW RETURNED	16-Aug-2016

GENERAL COMMENTS	this is an important study to nurses, the cross sectional survey used, the large sample size supported the result of your study. but it didn't reveal any new information to what already known. in page 3 line 24 you said that nurses are more likely to experience WPV but did not explain this information. id think that the range of sexual harassment is in some countries less than 13.02 % in line 29. the statistics you mention in line 39-49 are talking about physician you need to mention nurses statistics related WPV in china and world wide, and if there is previous study about nurses in china? if yes you need to say what is the contribution of your study, if not you need to mention it. in page 4 line 7 there are more important consequences than you reported, the method paragraph is not clear do you mean by- 80 nurses were extracted from department of emergency, neurology, obstetric and gynecology- does it mean 80 from all of these department or 80 from each department? in the discussion part you need to discuss more and compare your result with previous result in more clear way, you need to compare your results with previous result in china and neighbor countries or
---

	world wide to make clear the contribution and the uniqueness of your study. you need to give the reader the new information you found.
--	--

REVIEWER	Dr Teris Cheung School of Nursing, Hong Kong Polytechnic University, Hong Kong.
REVIEW RETURNED	12-Nov-2016

GENERAL COMMENTS	This large-scale cross-sectional study is recently conducted in China and the sample size is large. Nonetheless, international readers have little knowledge on the total no. of nurses currently working in China and which licensure board they are registered with. Readers should know whether this sample is a significant fraction representing the nursing population in China as a whole-though, it seems like authors were using stratified random sampling in their study. For the ethical consideration, authors mentioned about 'informed consent', how did they perform the 'informed consent' if it's a cross-sectional survey and the informed consent was conducted by 'whom'? Authors also mentioned that nurses were at high risk of WPV comparing with other healthcare counterparts but research design of this study cannot reach this conclusion because all the participants were nurses! Definitions of different types of WPV were unclear. It would be helpful to use the WHO definition of WPV in this study. Some references were a bit outdated. There are massive literatures on WPV internationally and globally. Authors also recommended training to mitigate the negative psychological effect of WPV but could you offer any examples of feasible interventions / training specific to nurse professionals? Is there any operational guidelines that may help to deter WPV in healthcare setting. Last but not least, in line 44, the funding session, authors said it's FOUNDED by blah blah blah, would it be 'FUNDED'? Please correct it.
---

VERSION 1 – AUTHOR RESPONSE

Respond to the Reviewer 1

Institution and Country: Azienda Ospedaliera di Perugia, Perugia (Italy)

Dear Professor Nicola,

Thank you very much for your valuable advice. We revised the manuscript according to your suggestion (The traces of change are represented in light blue). Modify as follows:

1. Competing Interests: None declared
2. We carry on the explanation to the “Yi Nao.” “Yi Nao” (where people employed by those in dispute with the hospital, together with family members of patients, took various measures to put pressure on the hospital, thereby profiteering).
3. We add the study limitations in the manuscript. The present study has several limitations. First, we collected data about whether nurses had experienced WPV over the past 12 months, so there may be recall bias in the results. Second, we studied tertiary and county-level hospitals, but we did not study specialized hospitals. However, this study might be effective in preventing WPV in general and public hospitals.

We wish you a happy work.

Best wishes,

Lei Shi

Department of Health Management, School of Public Health, Harbin Medical University, China

Respond to the Reviewer 2

Institution and Country: Jordan- balqaa applied university

Dear Professor Al-Rimawi,

Thank you very much for your valuable advice. We revise the manuscript according to your suggestion(The traces of change are represented in purple). Modify as follows :

1. Competing Interests: None declared
2. "The nurses are more likely to experience WPV" in page 3 line 24 (Manuscript for the first time). We add references and change to "Several substantial studies have suggested that nurses have a high risk of experiencing WPV.6-10 "
3. "range of sexual harassment" in page 3 line 29 (Manuscript for the first time). We added references and changed to "During the past 12 months, the incidence rate of physical violence for nurses in Ethiopia,7 South Korea,8 Jordan,9 Germany10 and Iran 3 ranged from 18.22% to 56.0%, verbal abuse from 63.8% to 89.58%, and sexual harassment from 4.7% to 19.7% "
4. "We did not mention nurses statistics related workplace violence in china and world wide" in page 3 line 39-49 (Manuscript for the first time).We add references and change to "Further, the incidence rate of WPV differs among nurses in various countries; for instance, the incidence was 76.0% in Greek, 82% in Pakistan, and 67% in Italy.5-6" ; "Most previous studies of WPV against in China have been conducted in the provinces and the samples are not sufficiently representative; they do not present an accurate picture of the incidence of WPV against nurses in Chinese general hospitals.18-20 "
5. "which may lead to the phenomenon that healthcare workers do not hope their children to be engaged in medical work."in page 4 line 7(Manuscript for the first time). We add references and change to "Moreover, violent incidents have a negative impact on the psychological welfare of the healthcare workers,23 who do not want their children to be engaged in healthcare.24 "
6. We give a detailed description of the sample extraction. "In order to select the same proportion of the workforce from each department, we sought to recruit 120 nurses from the Departments of Internal Medicine and Surgery, 80 nurses from the Departments of Emergency, Neurology, Obstetrics and Gynecology, and Pediatrics, and 40 nurses from the Departments of Stomatology, Ophthalmology, and Ear, Nose and Throat in each tertiary hospital. We distributed 10,560 questionnaires to 44 tertiary hospitals."
7. We compare our result with previous result in China and world wide. "Compared with previous Chinese studies,18-20 our study provides a comprehensive depiction of the incidence of WPV in Chinese comprehensive public hospitals. We have also documented the characteristics of the perpetrators and the coping style of nurses who experience WPV. This study demonstrates that WPV in China is higher than other countries. 34-37 "

We wish you a happy work.

Best wishes,

Lei Shi

Department of Health Management, School of Public Health, Harbin Medical University, China

Respond to the Reviewer 3

Institution and Country: School of Nursing, Hong Kong Polytechnic University, Hong Kong.

Dear Dr Cheung,

Thank you very much for your valuable advice. We revise the manuscript according to your suggestion(The traces of change are represented in green). Modify as follows :

1. Competing Interests: None declared
2. We increase the total number of nurses currently working in Chinese public hospitals and the number of registered nurses. We change to "In 2015, there were 3,069 general and public hospitals in China and approximately 920,700 registered nurses in public tertiary and county-level hospitals. We selected a sample of 21,360, approximately 2.30% of all nurses."
3. For the ethical consideration, we mentioned about "informed consent", "All participants gave informed consent to the researchers or to their head nurses before the survey".
4. We also mentioned that nurses were at high risk of WPV comparing with other healthcare counterparts but research design of this study cannot reach this conclusion. (The traces of change are represented in purple or green)We change to "Several substantial studies have suggested that nurses have a high risk of experiencing WPV.6-10 "
5. In this study, we redefine WPV according to the WHO definition of WPV. We change to "In this study, WPV is divided into physical violence and psychological violence, which includes verbal violence and sexual harassment in accordance with the definition of WPV in hospitals used by the World Health Organization (WHO), and the International Labour Office (ILO), and the specific situation in China. "
6. Some references were a bit outdated. We conducted a literature update.
 2. Alkorashy HA, Al Moalad FB. Workplace violence against nursing staff in a Saudi university hospital. *INT NURS REV* 2016;63(2):226–232.
 3. Fallahi K M, Oskouie F, Ghazanfari N, et al. The Frequency, Contributing and Preventive Factors of Harassment towards Health Professionals in Iran. *IJCBNM* 2015;3(3):156-164.
 5. Fafliora E, Bampalis V G, Zarlis G, et al. Workplace violence against nurses in three different Greek healthcare settings. *Work* 2015;53(3):551-560.
 6. Ferri P, Silvestri M, Artoni C, et al. Workplace violence in different settings and among various health professionals in an Italian general hospital: a cross-sectional study. *Psychology Research & Behavior Management* 2016;9:263-275.
 10. Schablon A, Zeh A, Wendeler D, et al. Frequency and consequences of violence and aggression towards employees in the German healthcare and welfare system: a cross-sectional study. *BMJ Open* 2011;2(5):421-421.
 15. Albashtawy M, Aljezawi M. Emergency nurses' perspective of workplace violence in Jordanian hospitals: A national survey. *Int Emerg Nurs* 2015;24:61-65.
 18. Zhao Y, Wang FM, Zhou CY, et al. Investigation and analysis of present situation of workplace violence at tertiary hospitals and county hospitals in Heilongjiang Province. *Medicine and Society* 2016;29(9):35-37.
 19. Yan JH, Hou J, Lin HQ, et al. Workplace violence in hospital setting to nursing staff and its influencing factors. *Journal of Nursing (China)* 2012;19(8A):4-7.
 23. Itzhaki M, Peles-Bortz A, Kostitsky H, et al. Exposure of mental health nurses to violence associated with job stress, life satisfaction, staff resilience, and post-traumatic growth. *International Journal of Mental Health Nursing* 2015;24(5):403–412.
 24. Yang JJ, Gao LL. A survey on willingness of nurses' children engage in medical work. *Education Teaching Forum* 2015;(3):73-74.
 35. Tiruneh BT, Biftu BB, Tumebo A A, et al. Prevalence of workplace violence in Northwest Ethiopia:

a multivariate analysis. BMC Nursing, 2016;15(1):1-6.

7. We also recommended training to mitigate the negative psychological effect of WPV . For instance, hospitals could provide violence-related training for nurses and provide post-WPV psychological support or a “debriefing room”, instruct all staff about the value of nurses and foster nurses’ pride in their work, develop an excellent hospital culture, etc.

There is no Chinese operational guidelines that may help to deter WPV in healthcare setting. But each hospital has some simple measures to reduce violence.

8. in line 44, the funding session. We change “founded” to “funded”.

We wish you a happy work.

VERSION 2 – REVIEW

REVIEWER	CHEUNG Teris School of Nursing, Hong Kong Polytechnic University, Hong Kong
REVIEW RETURNED	29-Dec-2016

GENERAL COMMENTS	1. Abstract requires further refinement, particularly the sentence structure;2. L37 – please rephrase ‘has increased gradually increasing over the few decades’3. L36-47 – if WPV towards nurses is the main focus of the study, authors should consider removing the prevalence of WPV towards doctors because in this manuscript, the prevalence of WPV towards doctors is not examined!4. Ethical consideration- authors seem to split the ethical approval from the research site in another paragraph. Please merge all the ethical approval / considerations under the same subtitle named ‘ethical considerations’.5. Authors suggested that training should provide to frontline nurses to prevent WPV. A good example to be utilized is the ‘Management of Violence’ and additionally, authors should critically re-think how to improve the psychological capital of these at risk nurses e.g. improving resilience to mitigate the negative impact brought by WPV.6. Lots of WPV preventive strategies have little empirical support in the discussion. There are lots of empirical evidence on anti-violence strategies in the literature. Please support your lines of discussion with recent empirical evidence to make your suggestions sound.
--

VERSION 2 – AUTHOR RESPONSE

Dear Dr Cheung,

Thank you very much for your valuable advice. We revised the manuscript according to your suggestion(The traces of change are represented in blue). Modify as follows :

1. Competing Interests: None declared

2. We further refine the abstract, especially the sentence structure. We change to “Objectives: The purpose of the present study was to explore the characteristics of workplace violence that Chinese nurses at tertiary and county-level hospitals encountered in the 12 months from December 2014 to January 2016, to identify and analyze risk factors for workplace violence, and to establish the basis for future preventive strategies.

Design: A cross-sectional study.

Setting: A total of 44 tertiary hospitals and 90 county-level hospitals in 16 provinces (municipalities or

autonomous regions) in China.

Methods: We employed stratified random sampling to collect data from December 2014 to January 2016. We distributed 21,360 questionnaires, and 15,970 participants provided valid data (effective response rate = 74.77%). We conducted binary logistic regression analyses on the risk factors for workplace violence among the nurses in our sample and analyzed the reasons for aggression.

Results: The prevalence of workplace violence was 65.8%; of this, 64.9% was verbal violence, and physical violence and sexual harassment accounted for 11.8% and 3.9%, respectively. Frequent workplace violence occurred primarily in emergency and pediatric departments. Respondents reported that patients' relatives were the main perpetrators in tertiary and county-level hospitals. Logistic regression analysis showed that respondents' age, department, years of experience, and direct contact with patients were common risk factors at different levels of hospitals.

Conclusions: Workplace violence is frequent in China's tertiary and county-level hospitals. The frequent occurrence of WPV in emergency and pediatric departments is also remarkable. It is necessary to cope with workplace violence by developing effective control strategies at individual, hospital, and national levels."

3.L37 – "has increased gradually increasing over the few decades" is changed to "In China, WPV in hospitals has increased gradually over the past few decades."

L36-47 –We removed the prevalence of WPV towards doctors. We changed to "According to the report from the Chinese Hospital Association, the proportion of hospitals experiencing WPV rose from 90% in 2008 to 96% in 2012 and the prevalence of sexual harassment has increased year by year.16 "

4. We merge all the ethical approval / considerations under the same subtitle named "ethical considerations".

ETHICAL CONSIDERATIONS

Ethical approval to undertake this study was granted by the Research Ethics Committee of Harbin Medical University in March 2014. We obtained consent from each hospital involved in the research processes. All participants gave informed consent to the researchers or to their head nurses before the survey, and participants' personal information was kept confidential.

5. We add the information about how to improve the psychological capital of these at risk nurses. We changed to "We also suggest improving psychological resilience for at-risk nurses to mitigate the negative impact of WPV and to prevent chronic diseases and reduce the incidence of mental illness.48 "

6.We add some recent empirical evidence on anti-violence strategies in the discussion. We changed to:

1) On the one hand, hospitals can establish a code green response team, comprising a charge nurse, security personnel, and primary nurse to manage the potentially violent situation. Dilman's study demonstrated that 85% of code green calls resulted in successful resolution of the violent incidents.42 Preventive measures for WPV among nurses include the following: increase awareness of potentially violent patients; wear suitable clothes; maintain proper positioning when communicating with patients; keep a safe distance; maintain the correct posture; and listen actively. A study by Hill et al. showed a 65% reduction in staff injuries, from 2.2 per week to 0.77 per week, during the 1-year intervention period.43

2) Stievano et al.'s study has shown that patient care can be affected when nurses are not respected.44

3) Previous studies have shown that strengthening hospital management can improve patient satisfaction.45

4) Fine safety management can significantly improve the quality of nursing and patients' satisfaction, and reduce conflict.46

5) If nurses have the support of managers, they may be more willing to consult psychologists, psychiatrists, and mental health professionals to identify and treat any sudden disorder.47

6) Nurses should be familiar with all the resources in their work and family communities for solving any problems and should keep themselves in good mental health.47

References

42. Dilman Y. Code green for workplace violence. *Critical Care Nurse* 2015;35:34-35.
43. Hill AK, Lind MA, Tucker D, et al. Measurable results: Reducing staff injuries on a specialty psychiatric unit for patients with developmental disabilities. *Work (Reading, Mass.)* 2015;51(1): 99-111.
44. Stievano A, Bellass S, Rocco G, et al. Nursing's professional respect as experienced by hospital and community nurses. *Nursing Ethics* 2016.
45. Al Fraihi KJ, Latif SA. Evaluation of Outpatient Service Quality in Eastern Saudi Arabia. Patient's expectations and perceptions. *Saudi Medical Journal* 2016;37(4):420-428.
46. Mei XM. Application of Fine Management in Safety Management in Geriatric Ward. *Hospital Management Forum* 2016.
47. Teris C, Yip PSF. Depression, anxiety and symptoms of stress among Hong Kong nurses: a cross-sectional study. *Inter J Env Res Pub Heal* 2015;12(9):11072-11100.
48. Teris C, Yip PSF. Lifestyle and Depression among Hong Kong Nurses. *Inter J Env Res Pub Heal*, 2016;13(1):135.

We wish you a happy New Year.